# Two–Dimensional and Doppler trans-thoracic echocardiographic patterns of suspected pediatric heart diseases at Tibebe—Ghion specialized Teaching Hospital and Adinas General Hospital, Bahir Dar, North-west Ethiopia:–An experience from an LMIC

**Tesfaye Taye Gelaw**[1]*, **Amare Aschalew Yehuala**[1], **Senay Zerihun Mengste**[1], **Yalemwork Anteneh Yimer**[1], **Habtamu Bayih Engida**[2], **Abiot Tefera Alem**[2]

**1** Department of Pediatrics and Child Health, College of Medicine and Health Sciences, Bahir Dar University, Bahir Dar, Ethiopia, **2** Department of Internal Medicine, College of Medicine and Health Sciences, Bahir Dar University, Bahir Dar, Ethiopia

* ttgela2020@gmail.com

## Abstract

### Background

Transthoracic Echocardiography is the first-line, non-invasive, and accessible imaging modality to evaluate heart disease anatomy, physiology, and hemodynamics. We aim to describe the trans-thoracic echocardiography pattern of pediatric heart diseases and reasons for referral in children referred to Bahir Dar University Tibebe–Ghion Hospital and Adinas General Hospital.

### Method

A descriptive cross-sectional study of the archived Transthoracic, Two Dimensional, and Doppler Echocardiography assessments of children from birth to fifteen years of age performed between June 2019 to May 2023 was done. Data were collected retrospectively from February 01, 2023 –May 31, 2023. Categorical variables like gender, referral reasons for echocardiography, and patterns of pediatric heart lesions were analyzed in the form of proportions and presented in tables and figures. Discrete variables including age were summarized as means (SD) and medians(IQR).

### Results

Out of 3,647 Children enrolled; 1,917 (52.6%) were males and 1,730 (47.4%) were females. The median (IQR) age of children enrolled was 24 months (5 to 96). Cardiac murmur (33%) was the most common reason for echocardiography followed by, Respiratory Distress (18%), Syndromic Child (15%), easy fatigability/ Diaphoresis (14.3%), congestive heart failure (14%), and rheumatic fever (13.2%). Congenital heart defect (CHD) accounts for 70% of

**Data Availability Statement:** All relevant data are within the manuscript and its Supporting Information files.

**Funding:** The author(s) received no specific funding for this work.

**Competing interests:** The authors have declared that no competing interests exist.

**Abbreviations:** ASD, Atrial septal defect; AVSD, Atrioventricular septal defect; CHD, Congenital Heart Defect; DORV, Double outlet right ventricle; LVOTO, Left ventricular outflow tract obstruction; MV, Mitral valve; PDA, Patent ductus arteriosus; PHT, Pulmonary Hypertension; PPHTN, Persistent Pulmonary Hypertension of the newborn; PS, Pulmonary stenosis; TGA, Transposition of great arteries; TOF, Tetralogy of Fallot; VSD, Ventricular septal defect.

all heart diseases, followed by rheumatic heart disease (21%). Isolated ventricular septal defect(VSD) was the most common CHD (21%) followed by isolated Patent ductus arteriosus (15%), isolated atrial septal defect (10%), Isolated atrioventricular septal defect (6%) and isolated pulmonary stenosis (5%). Cyanotic CHD accounts for 11.5% of all heart diseases. Tetralogy of Fallot (30%), d-TGA (20%), and double outlet right ventricle (19%) were the most common cyanotic CHDs.

## Conclusions

In our study, congenital heart lesions are the most common diagnosis and cardiac murmurs are the most common presenting reasons for echocardiography evaluation.

## Introduction

According to the WHO Report, across all age groups; Cardiovascular diseases account for the most Non-Communicable Disease (NCD) deaths, or 17.9 million people annually [1]. With the United Nations' adoption of the Sustainable Development Goals; specifically Sustainable Development Goal 3.2, that aims to end preventable deaths of newborns and children under 5 years of age, a significant reduction has been achieved in global mortality from communicable disease [2, 3].

It is notable that Pediatric Heart diseases including Congenital Heart Diseases (CHD) and acquired heart diseases (AHD), which affect a significant number of children, have largely been left untouched in Low–and Middle–low-income countries in contrary to SDG 3.4, which aims to reduce by one-third premature mortality from non-communicable diseases [4]. Unorganized referral system, treatment and follow-up gaps; and Lack of availability of diagnostic equipment, infrastructure, and trained personnel are some of the main barriers to providing quality Pediatric Heart Disease care in low–and Middle–middle-income countries [5]. Therefore, Screening, Detection, and treatment of NCDs, including Heart diseases, are key components of the response to NCDs [1].

Transthoracic Echocardiography is the first-line, non-invasive, portable, and accessible imaging modality in the diagnosis and evaluation of pediatric heart disease anatomy, physiology, and hemodynamics [6–10]. Given the increasing cost of healthcare and limited human power, it is essential to ensure that the use of cardiovascular imaging is appropriate [11].

Reasons for Pediatric echocardiogram encompass a wide range of clinical conditions. These include: respiratory distress, congestive heart failure, cyanosis, exertional chest pain, syncope, failure to thrive, murmurs, cardiomegaly, Infective endocarditis, rheumatic fever, thromboembolic events, arrhythmia, systemic hypertension, pulmonary hypertension, syndromes associated with cardiovascular disease, cardiogenic shock, heart surgery, and Fetal echocardiography [6, 12–17].

So far, there is limited information regarding echocardiographic patterns and the reason for referral for echocardiography of suspected pediatric cardiac patients with such a larger number in Ethiopia [18]. Having such a piece of descriptive detailed information from a tertiary health care facility would provide very relevant information to further study the pattern and nature of pediatric heart diseases in other corners of the country. It also will complement the available information on pediatric echocardiographic studies at the global level from the LMIC's side.

Cognizant of this, we undertook a retrospective review of four-year archived Trans-Thoracic echocardiography done using the segmental analysis approach for Pediatric patients referred for echocardiography to the cardiac care units. This study will help to define the pattern of Pediatric cardiac disease in our community, and further add to the national and global database.

## Method

### Setting and participants

The study was conducted on 3,647 pediatric clients with complete documentation and no previous echocardiography assessment who were referred to the Paediatric Echocardiography laboratories of Adinas General Hospital; a privately owned Hospital and Bahir Dar University Tibebe-Ghion Teaching Hospital, from June 2019 to May 2023. The Hospitals serve as referral centers for pediatric cardiac diagnostics and medical treatment for Northwest and part of West Ethiopia. Part of the clients who are residing near these two hospitals were being cared for in the respective hospitals and the rest majority were being referred back to the referring health institutes with complements for follow-up and medical management in their vicinity. Only a few of them were referred to a center where surgical and transcatheter interventions are available due to economic reasons. Data were collected from February 01, 2023 –May 31, 2023, on a retrospective basis from the archives of the respective Hospitals'. An echocardiography study was performed by a pediatric Cardiologist and doubtful findings were discussed amongst the cardiologists before writing an official report. Data were collected by Trained Pediatricians and adult and pediatric Cardiologists.

### Study tool

Phillips iE33 xMATRIX model ultrasound system echocardiography was used for collecting echocardiographic data from Bahir Dar University Tibebe-Ghion Hospital and the GE LOGIQ C5 Premium model was used for echocardiographic data collection from Adinas General Hospital. 2D echocardiography and Doppler with all cardiac measurements were taken. Echocardiography evaluation was made using the segmental sequential analysis [8].

### Study design

Descriptive cross-sectional analysis of all archived completely documented echocardiography assessment reports done using sequential segmental analysis was done [6, 19–21]. Descriptive analysis was performed for age, gender, reasons for referral for echocardiography, and echocardiographic diagnoses. Ethical clearance and waiver of consent were obtained from the ethics and research committee of the College of Medicine and Health Sciences of Bahir Dar University with protocol number: 795/2023.

### Diagnosis and classification

**Clinically suspected Myocarditis [22–24].**   **Acute:** A poorly functioning ventricle with or without dilation, recent heart failure symptoms, and viral infectious symptoms in the preceding weeks.

**Fulminant:** Presents as a cardiogenic shock; tachyarrhythmias are common, and inotropic or mechanical circulatory support (MCS) may be needed.

**Cardiomyopathy** [25, 26]: Structural and functional abnormalities of the ventricular myocardium that are unexplained by flow-limiting coronary artery disease or abnormal loading conditions.

**Primary:** The heart is the only involved organ. Can be Genetic, Non-Genetic, or Acquired.

**Secondary:** A manifestation of a systemic disorder. shows pathological myocardial involvement as part of a large number and variety of generalized systemic disorders.

## Statistical analysis

Recorded data was cleared, coded, and entered into SPSS version 25 for analysis. Categorical variables including gender, reasons for referral for echocardiography, and diagnostic patterns of pediatric heart lesions were analyzed in the form of proportions and percentages and presented in tables and figures. Discrete variables including age were analyzed and summarized as means (SD) and medians (IQR).

## Result

3,672 Pediatric clients suspected of cardiac lesions visited Bahir Dar University Tibebe–Ghion Hospital and Adinas General Hospital echocardiography Laboratories for the first time from June 2019 –to May 2023. Of which twenty-five were excluded because of incomplete documentation and the remaining 3,647 were subjected to analysis (Fig 1).

Out of the 3,647 Pediatric echocardiography first-time assessments with complete documentation; 501 (13.7%) were neonates, 916 (25.1%) were infants, 389 (10.7%) were toddlers,

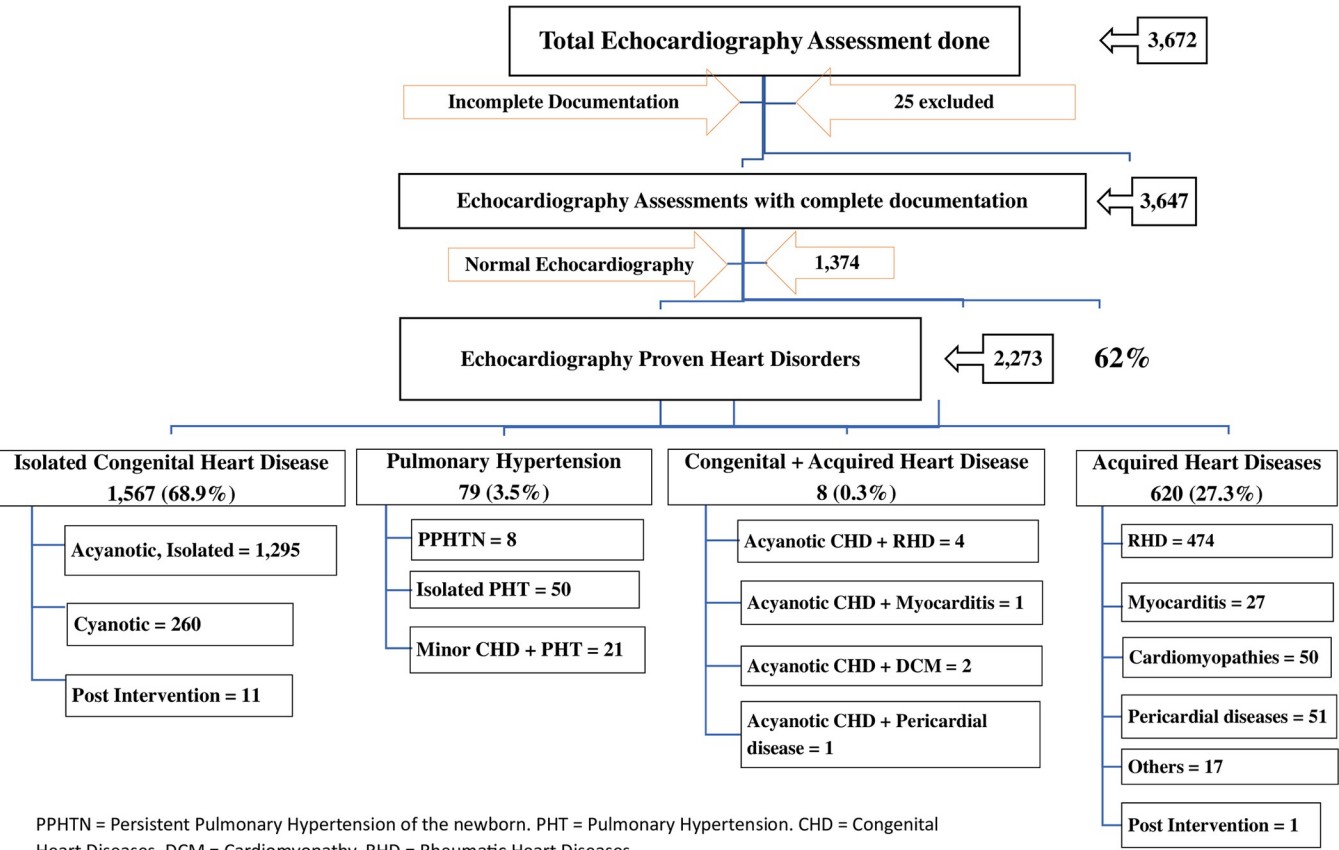

**Fig 1. Echocardiography study flow diagram: Bahir Dar University Tibebe-Ghion and Adinas Hospitals, June 2019—May 2023.** PPHTN = Persistent Pulmonary Hypertension of the newborn. PHT = Pulmonary Hypertension. CHD = Congenital Heart Diseases. DCM = Cardiomyopathy. RHD = Rheumatic Heart Diseases.

627 (17.2%) were preschoolers, 761 (20.9%) were school age, and 453 (12.4%) were adolescents. The median (IQR) age of children subjected to the study was 24 (5–96) months. There were 1,917 (52.6%) males and 1,730 (47.4%) females, making a male-to-female ratio of 1.1:1.

The most common reason for referral for echocardiography was cardiac murmur (33%) followed by Respiratory Distress (18%), Syndromic Child (15%), easy fatigability (14.3%), Congestive Heart Failure (14%), Rheumatic Fever (acute & recurrent) (13.2%), dyspnea on exertion (7.5%), recurrent chest infection (7%), cyanosis (5.8%), palpitation (5.6%), and Others (Fig 2.).

Out of the 3, 647 pediatric clients with complete documentation and first-time visits; 2,273 children had one or more cardiac lesions on echocardiography, reflecting the proportion of children identified with cardiac lesions to be 62%. The remaining 38% were found to be normal while having one or more of the clinical manifestations of cardiac lesions as a reason for referral. All children presenting with cyanosis, clubbing, Pericardial friction rub, and thrombo-embolic events had cardiac lesions identified using echocardiography evaluation; whereas pediatric clients whose reason for referral was being infant of diabetic mother

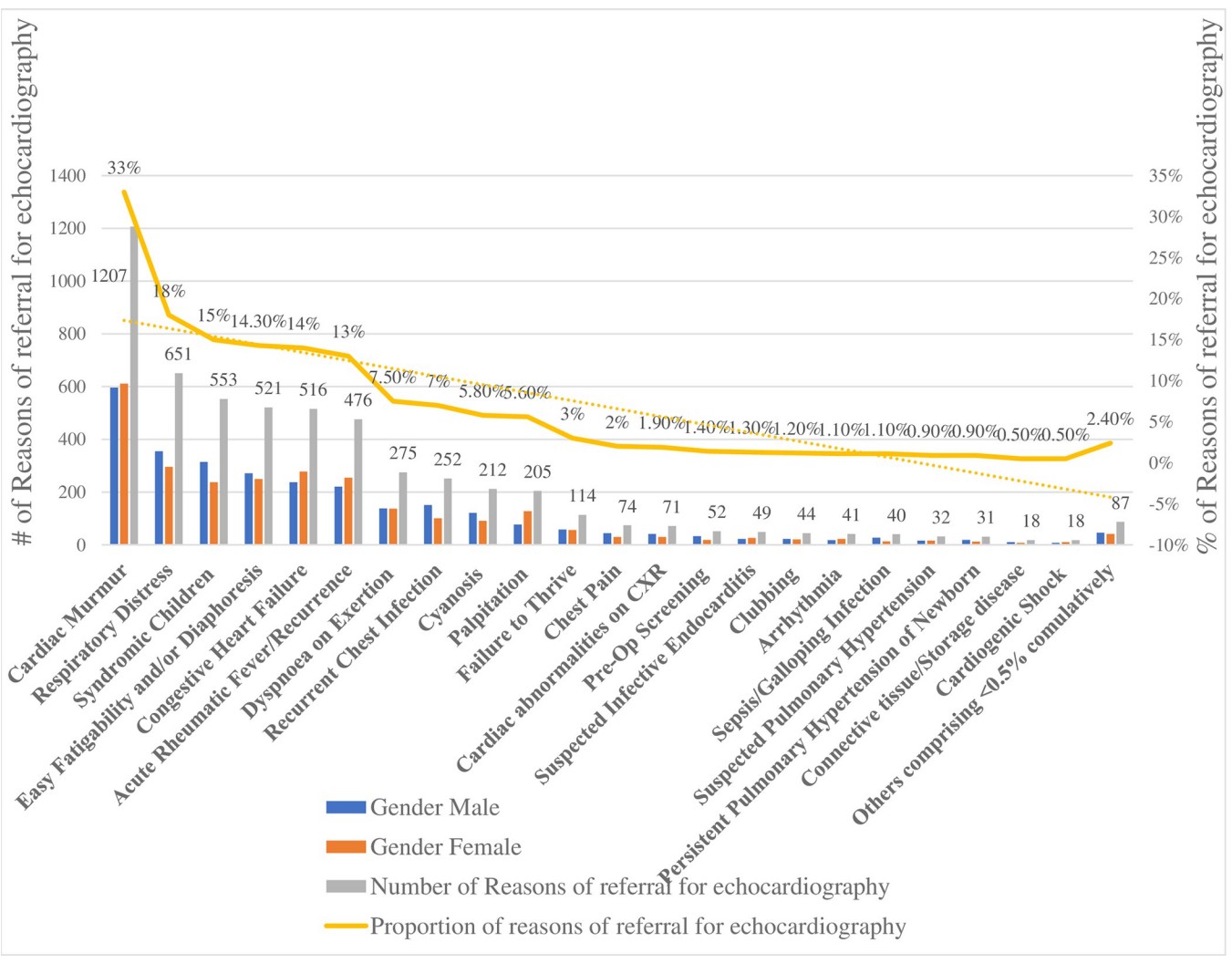

**Fig 2. Reasons for referral for echocardiography: Bahir Dar University Tibebe-Ghion and Adinas Hospitals, June 2019—May 2023.**

(16.7%), Syncope (18.2%), suspected Persistent pulmonary Hypertension of the newborn (19.4%), recurrent chest infection (19.8%), congenital stridor (20%) and non–selective Pre-Operative screening (21.2%) were having lower proportion of heart lesions to uncover any cardiac abnormalities (S1 Table).

Congenital heart defects (CHDs) account for 1597(70%) of all Pediatric Heart diseases; followed by rheumatic heart diseases, 479(21%); Pulmonary hypertension not caused by left side heart failure, 79(3.5%); Pericardial diseases, 51(2.2%); all forms of cardiomyopathies, 52(2.3%); and clinically suspected myocarditis, 28(1.2%). Isolated ventricular septal defect(VSD) was the most common CHD (21%) followed by isolated Patent ductus arteriosus (15%), isolated atrial septal defect (10%), Isolated atrioventricular septal defect (6%) and isolated pulmonary stenosis (5%). Tetralogy of Fallot is the leading cyanotic congenital heart defect accounting for 3.4% of all heart diseases (4.9% of all CHDs and 30% of all cyanotic CHDs) followed by d-TGA (2.2% of all heart diseases, 3.3% of all CHDs and 19.5% of all cyanotic CHDs), and double outlet right ventricle (2.2% of all heart diseases, 3% of all CHDs and 19% of all cyanotic CHDs) (Table 1).

## Discussion

Our study shows very important information regarding the reasons for referral, trends, and patterns of heart diseases in children. Among children referred for echocardiography, 62% had heart lesions which is in line with an Indian study [27]. In our study, the most commonly reported reason for referral for echocardiography is detecting murmur on clinical evaluation (33%) as reported in different parts of the world [28, 29]. This finding provides insight into the various reasons for referral for echocardiography evaluation provided by all categories of health care providers. It also encourages healthcare providers to advocate and practice routine clinical evaluation of children during a scheduled or unscheduled health service visit.

Our study also showed that infants from one to twelve months of age have the highest proportion of heart defects compared to the other age groups (27.1%). According to Zimmerman's work on CHDs in low-and-middle-income countries, it is more common for children to present in later stages of the disease and at older ages in low-resource settings [30]. This might be, in part, explained by delayed presentation and diagnosis fueled by an unorganized referral system, a lower level of awareness and health-seeking behavior; and a Lack of availability of diagnostic equipment, infrastructure, and trained personnel in the vicinity. In addition, newborns with critical CHD may be misdiagnosed or may not make it to the hospital and die at home or somewhere else which in turn will decrease the proportion of newborns with CHD in our study.

Congenital heart defect is the most common heart lesion in our study. This finding is not in line with what was previously thought about the proportion of congenital heart defects in low-and middle-income countries [31, 32]. The possible explanation is that, though incidence and prevalence can be affected by genetic and environmental factors, there is no strong clue that CHDs are less in LMICs compared to HICs. In addition, the scarcity of diagnostic services in the former days in these regions has led to undermining the caseload and possible misdiagnosis.

The most common congenital heart defect identified in our study is isolated ventricular septal defect followed by isolated patent ductus arteriosus and isolated atrial septal defect respectively. This is similar to most other studies done in both developed and developing countries except for the Iranian and Nigerian studies that reflect atrial septal defect is the most common one [28, 29, 33–39]. This difference from the Iranian and Nigerian studies is possible because children with atrial septal defects are asymptomatic and hence are less likely to show

**Table 1. Relative frequency of paediatric heart diseases: Tibebe-Ghion and Adinas Hospitals, June 2019 –May 2023.**

|  | 0–28 days | 1–12 months | ≥1–2 years | ≥2–6 years | ≥6–12 years | ≥12–18 years | Total | Percent |
|---|---|---|---|---|---|---|---|---|
| **Acyanotic CHDs, Isolated** | **241** | **461** | **184** | **229** | **119** | **61** | **1295** | **57** |
| Isolated ASD | 47 | 47 | 18 | 22 | 14 | 10 | 158 | 12.2 |
| Isolated VSD | 36 | 122 | 54 | 68 | 35 | 15 | 330 | 25.5 |
| Isolated AVSD | 9 | 44 | 17 | 16 | 7 | 3 | 96 | 7.4 |
| Isolated PDA | 50 | 79 | 37 | 44 | 17 | 7 | 234 | 18.1 |
| Isolated Pulmonary Stenosis | 10 | 31 | 13 | 17 | 8 | 4 | 83 | 6.4 |
| Two or more combined acyanotic CHDs | 75 | 124 | 38 | 49 | 26 | 8 | 320 | 24.7 |
| Ebstein anomaly of Tricuspid valve | 3 | 1 | 0 | 2 | 2 | 4 | 12 | 0.9 |
| Other acyanotic CHDs | 11 | 13 | 7 | 11 | 10 | 10 | 62 | 4.8 |
| **Minor CHD + Severe PHT** | **0** | **13** | **6** | **1** | **0** | **1** | **21** | **0.9** |
| **Acyanotic CHD + Acquired heart disease** | **0** | **1** | **1** | **2** | **4** | **0** | **8** | **0.35** |
| **S/P Surgical or device intervention** | **0** | **0** | **1** | **5** | **4** | **0** | **10** | **0.4** |
| **PHT, (Isolated + PPHTN)** | **9** | **15** | **12** | **16** | **2** | **4** | **58** | **2.55** |
| **Cyanotic Congenital heart defects** | **37** | **103** | **34** | **54** | **23** | **10** | **261** | **11.5** |
| Tetralogy of Fallot | 7 | 23 | 12 | 17 | 13 | 6 | 78 | 29.9 |
| d-TGA | 11 | 23 | 4 | 10 | 3 | 0 | 51 | 19.5 |
| d-TGA + Intact IVS/Restrictive VSD | 3 | 9 | 0 | 0 | 0 | 0 | 12 | |
| d- TGA + adequate mixing | 7 | 9 | 3 | 8 | 1 | 0 | 28 | |
| d-TGA + VSD + LVOTO | 1 | 5 | 1 | 2 | 2 | 0 | 11 | |
| Double Outlet Right Ventricle/DORV/ | 6 | 22 | 9 | 9 | 1 | 2 | 49 | 18.8 |
| DORV—VSD type | 0 | 7 | 4 | 2 | 1 | 0 | 14 | |
| DORV-TOF type | 1 | 1 | 1 | 2 | 0 | 1 | 6 | |
| DORV-TGA type | 0 | 4 | 0 | 0 | 0 | 0 | 4 | |
| DORV with Non-committed VSD | 3 | 5 | 3 | 2 | 0 | 1 | 14 | |
| DORV-AVSD-PS-Hetrotaxy | 2 | 4 | 1 | 2 | 0 | 0 | 9 | |
| DORV with no VSD | 0 | 1 | 0 | 1 | 0 | 0 | 2 | |
| Double Outlet Left Ventricle | 2 | 0 | 1 | 0 | 2 | 1 | 6 | 2.3 |
| Tricuspid Atresia | 5 | 15 | 3 | 6 | 2 | | 31 | 11.9 |
| Truncus Arteriosus | 2 | 8 | 2 | 5 | 2 | | 19 | 7.3 |
| Hypoplastic Left Heart Syndrome | 2 | 8 | 0 | 0 | 0 | | 10 | 3.8 |
| Mitral atresia | 1 | 2 | 1 | 3 | 0 | | 7 | 2.7 |
| Other cyanotic CHDs | 1 | 2 | 2 | 3 | 0 | 1 | 9 | 3.4 |
| S/P Arterial switch operation | 0 | 0 | 0 | 1 | 0 | 0 | 1 | 0.4 |
| **Acquired Heart Diseases** | **2** | **24** | **16** | **79** | **274** | **225** | **620** | **27.3** |
| Rheumatic Heart Diseases | 0 | 0 | 0 | 45 | 240 | 189 | 474 | 76.5 |
| Myocarditis | 1 | 11 | 5 | 5 | 4 | 1 | 27 | 4.4 |
| Cardiomyopathies | 1 | 10 | 7 | 14 | 10 | 8 | 50 | 8 |
| Pericardial Diseases | 0 | 2 | 4 | 10 | 17 | 18 | 51 | 8.2 |
| Other Acquired Heart Diseases | 0 | 1 | 0 | 5 | 3 | 8 | 17 | 2.7 |
| S/P MV Commissurotomy | 0 | 0 | 0 | 0 | 0 | 1 | 1 | 0.2 |

to health institutes with clinical features of heart diseases, unlike children who were screened at school and in the community like in the case of the Nigerian study. The Iranian study focused only on neonates in whom small atrial septal defects are expected to close and decrease the proportion of the defect in later ages and lack generalizability [39, 40].

In our study, Tetralogy of Fallot is the leading cyanotic congenital heart defect followed by d-TGA, and double outlet right ventricle respectively. These findings are consistent with

studies done in other sites [41–44]. Ventricular septal defect and tetralogy of Fallot were the most common acyanotic and cyanotic lesions, respectively. These findings are similar to studies done elsewhere [28, 32, 36, 43].

Rheumatic heart disease is the leading acquired heart disease in our study followed by cardiomyopathies and/or clinically Suspected Myocarditis and Pericardial diseases respectively. This finding is similar to studies done in Nigeria and other global data in general [31, 45]. Other data from the USA and Japanese community showed the most common acquired heart disease to be Kawasaki Disease [46]. This difference may be explained by environmental, genetic, and socio-cultural differences. We also had children with both CHD and acquired heart diseases. These clients were evaluated rigorously to substantiate their diagnosis using operational definitions. Diagnosing acquired heart disease while the child is having a congenital heart lesion is challenging as there are no clear criteria set to diagnose such conditions in children.

## Conclusion

Congenital heart defects were the most common echocardiographic diagnosis, and the presence of a murmur was the most common reason for referral for echocardiography assessment. VSD, PDA, and ASD were the most common acyanotic heart defects while Tetralogy of Fallot and Double outlet right ventricle topped the cyanotic defects.

## Limitations

This study is a descriptive retrospective institutional study. Hence, generalizability is limited as it doesn't represent the whole population and is only descriptive. This study also doesn't address the appropriateness of "not performing" echocardiography; given children are presented to the respective units for echocardiography assessment. Diagnosing CHDs combined with acquired heart diseases using transthoracic echocardiography in children has no set criteria so far inciting the need for future work.

## Recommendations

Additional studies are warranted to further define the pattern of pediatric cardiac disease in our environment, and add relevant information to the national and global database representing LMICs.

## Supporting information

**S1 Table. Proportion of children with 1.**
(DOCX)

**S1 File.**
(ZIP)

**S2 File.**
(ZIP)

## Acknowledgments

We thank the Pediatric and Child health residents and Clinical nurses at the cardiac unit of Bahir Dar University Tibebe–Ghion Hospital for their relentless support of the pediatric cardiac services being provided at the unit. Our thanks also go to the Administration and Radiology Unit of Adinas General Hospital for providing uswith the necessary documents.

## Author Contributions

**Conceptualization:** Tesfaye Taye Gelaw.

**Data curation:** Tesfaye Taye Gelaw.

**Formal analysis:** Tesfaye Taye Gelaw, Amare Aschalew Yehuala, Abiot Tefera Alem.

**Investigation:** Tesfaye Taye Gelaw.

**Project administration:** Tesfaye Taye Gelaw, Senay Zerihun Mengste, Habtamu Bayih Engida.

**Supervision:** Tesfaye Taye Gelaw, Yalemwork Anteneh Yimer.

**Writing – original draft:** Tesfaye Taye Gelaw.

**Writing – review & editing:** Tesfaye Taye Gelaw, Amare Aschalew Yehuala, Senay Zerihun Mengste, Yalemwork Anteneh Yimer, Habtamu Bayih Engida, Abiot Tefera Alem.

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
