## [Decision Letter · Decision Letter 0]

1 Dec 2023

PONE-D-23-28984Two – Dimensional and Doppler Transthoracic Echocardiographic Patterns of suspected Paediatric Heart Diseases at Tibebe – Ghion Specialized Teaching Hospital and Adinas General Hospital, Bahir Dar, North West Ethiopia: – an experience from a LMIC.PLOS ONE

Dear Dr. Gelaw,

Thank you for submitting your manuscript to PLOS ONE. After careful consideration, we feel that it has merit but does not fully meet PLOS ONE’s publication criteria as it currently stands. Reviewers' comments are appended at the bottom of this email. Therefore, we invite you to submit a revised version of the manuscript that addresses the points raised during the review process. In my opinion, your manuscript should focus on a single clear objective. If you are reporting the spectrum of pediatric cardiac conditions you see in your practice, this should guide your methodology, results and discussion sections. If it's something else, it has to be spelled correctly. Additionally, presentation of results has to be concise and summarized. 

We look forward to receiving your revised manuscript.

Kind regards,

Academic Editor

PLOS ONE

- 10.1016/j.echo.2006.09.001

- 10.4103/jiae.jiae_60_20

In your revision ensure you cite all your sources (including your own works), and quote or rephrase any duplicated text outside the methods section. Further consideration is dependent on these concerns being addressed.

4. We are unable to open your Supporting Information file [Echo Submited to PLOS ONE.rar]. Please kindly revise as necessary and re-upload.

Reviewers' comments:

Reviewer's Responses to Questions

**Comments to the Author**

1. Is the manuscript technically sound, and do the data support the conclusions?

Reviewer #1: Partly

Reviewer #2: Partly

2. Has the statistical analysis been performed appropriately and rigorously? 

Reviewer #1: Yes

Reviewer #2: No

3. Have the authors made all data underlying the findings in their manuscript fully available?

Reviewer #1: Yes

Reviewer #2: No

4. Is the manuscript presented in an intelligible fashion and written in standard English?

Reviewer #1: No

Reviewer #2: No

5. Review Comments to the Author

Reviewer #1: Echocardiographic patterns

I would like to thank the authors for doing such a great job. I have the following comments/concerns:

Major concern:

- The aim of the study is not clear; is it to assess the “ Status of appropriate Use Criteria of Echocardiography in outpatient department of the study setting” or is it to determine “The magnitude of cardiac diseases among the screened pediatric patients in the study setting” or is it to assess “Correlation between clinical symptoms and echocardiographic covariates”. It is good if the authors can address only one of the above mentioned study aim since I saw authors using the above mentioned “aims” interchangeably in the manuscript.

Other comment:

- The title is lengthy.; better if you shorten it and also make it reflect the aim/finding of the study.

- There are punctuation errors and some grammar errors as well. E.g The authors are using the semicolon punctuation mark when they are listing items

Abstract Section:

- The result section in the abstract is too long.

- Conclusion in the abstract reads as “Indication based pediatric echocardiography has shown higher disease detection.” The basis of this conclusion is not clear. Which result does support such finding and what does it imply, does it mean the clinicians are following the “appropriate Use criteria”?

Introduction section:

- I recommend the authors either to remove or modify paragraph-4 on page-3 and paragraph -2 on page – 4 since they are not important to explain why the study is important.

- At the end of the introduction section the authors have stated that “..the study will help to define the pattern of pediatric cardiac disease…”. But in the abstract section, the authors stated “… we investigated indications for transthoracic echocardiography…”. Please be clear with aim of the study.

Methods section:

- It is good if the authors can provide information regarding data collection technique. E.g Were the data collected from the chart or from the saved video clips; who collected the data (physician, cardiologist….); who did the ECHO (cardiologist…), How was doubtful finding resolved….?

- Why the authors included post intervention cases in the study is not clear.

Result section:

- “3,672 clients had a visit to…” Does this mean all the pediatric patients who visited the hospital or…?

- I recommend the authors to remove some of the figures/tables so that the result section will be precise and concise in order to make it appealing for readers. For example, think about the importance of Table-2 and Table-3. Also, Table 4-2 appears that it is continuation of Table 4-1. The rows and columns of the two tables are too many and it is good if you can merge some variables and shorten it.

- It is good if the authors can cross-check the journal’s recommendation on how to write table headings.

- Title of Table -1 is not clear; what does “Diagnostic yield of TTE” means? To my understanding, diagnostic “yield of TTE” is the number of cardiac patients detected by the TEE and who subsequently proven to have the cardiac disease by other standard test divided by the total number of patients who underwent the screening. But Table-1 is not depicting such data.

Discussion section:

- I believe that the authors can improve the discussion section by focusing on the major findings of the study and the implication of the findings. As it stands right now, the discussion section mainly focuses on comparing prevalence values across different settings. That makes the discussion very superficial.

- For example, the authors have found out that screened infants have more prevalence of cardiac disease than other age groups. What does this imply? Is it because kids with cardiac disease die early? If so substantiate with evidence.

- Also, why CHD is more common than acquired heart disease among the screened population in a setting where RHD is assumed to be very prevalent?

- What does the 62% correlation between clinical suspicion of cardiac disease and ECHO finding of the disease indicate? Does it show the clinicians are good in assessing cardiac disease or vice versa? Or is there other explanation?

Reviewer #2: Topic: ‘Two – Dimensional and Doppler Trans-Thoracic Echocardiographic Patterns of suspected Paediatric Heart Diseases at Tibebe – Ghion Specialized Teaching Hospital and Adinas General Hospital, Bahir Dar, North West Ethiopia: – an experience from a LMIC’.

Authors: Tesfaye Taye Gelaw, Habtamu Bayih Engida, Abiot Tefra Alem, Amare Aschalew Yehuala, Senay Zerihun Mengste, Yalemwork Anteneh Yimer

Journal: PLOS ONE

Version 1

Date: Dec 1,2023

Dear Authors,

Thanks for bringing this important public health issue. It was an excellent initiative to investigate common indications for pediatric echocardiography and also reporting common findings.

Here below you find comments and suggestions that will help to refine your manuscript.

Abstract:

- Good.

- Please check the indications percentage as it adds up above 100% (133.4%).

- Please check and reconcile what you say detection rate. Detection rate is used when a test is compared with a gold standard (sensitivity in statistics). I don’t think you wanted to say that from what can be seen from the results in the manuscript. Here your gold standard is Echo which is compared against clinical symptoms/signs indications. Here it seems you are reporting the positive predictive value (PPV) of a given sign/symptom for picking a cardiac disease in children (as it is confirmed by Echo)

- You also reported a univariate analysis output in percentages. I don’t see here any report of statistical method in terms of P value or others tests.

Introduction:

It is too long. Please try to shorten this section.

Methods:

Settings: Please mention here the details of echocardiography machines that used (brand, model, probes, image storage capacity and access methods etc) for diagnostic purpose. Mention also here the capacity of the hospitals in terms of treatment, care and follow up of these diagnosed patients.

Analysis: Univariate analysis mentioned but no report of any statistical outputs like P value.

Please include ethical approving institution with its minutes.

In operational definition it is clear that you used the 2012 WHF criteria for RHD diagnosis given the time of your study. Please stress this in the discussion section of your paper as we have new 2023 WHF RHD criteria where borderline and other names have been removed. That way you will avoid confusion as publication of your paper if accepted will be after the new 2023 WHF guideline.

Results:

Figure 1 is good but try to make it clearer in relation to the numbers and percentages.

Page 6 indications add up above 100%.

The figures and tables are good. You have too much information and try to limit what you want to send to the readers. It looks you are reporting the positive predictive value of symptoms/signs for the diagnosis of heart disease in children. You are also reporting the common echocardiographic abnormalities in children.

You have detailed descriptive of echocardiographic abnormalities.

You may do univariate analysis and report the association between symptoms/signs and heart disease in children. That will make your study more clinically relevant.

Page 10 Tables 3-1 and 3-2 may be Tables 4-1 and 4-2. Please check that.

Discussion, conclusion and recommendations:

Please check and decide how you want to present the paper (PPV vs association between symptoms/signs and heart disease). You may revise this section and the title after that.

References: Ok

Additional comment: Please check syntax, capitalization and other minor errors in your manuscript for easy readability. Your paper can be revised after you decide what it needs to send to the readers.

6. PLOS authors have the option to publish the peer review history of their article (what does this mean?). If published, this will include your full peer review and any attached files.

Reviewer #1: **Yes: **Atnafu Mekonnen Tekleab

Reviewer #2: No

---

## [Author Response · Author response to Decision Letter 0]

3 Jan 2024

Response to academic editor’s and reviewers’ comments

1. Responses to academic editor’s comment/Journal requirements:

1.1. Please ensure that your manuscript meets PLOS ONE's style requirements, including those for file naming. 

Thanks for the recommendations provided to abide with the PLOS ONE’s style requirements. The manuscript is revised and edited in such a way that suits with the requirements.

1.2. We noticed you have some minor occurrence of overlapping text with the following previous publication(s).

Yes it overlaps with the publications mentioned (probably, it might be because of repetitive referencing of the studies in our day to day clinical practice). Thanks for the in depth evaluation of our manuscript. We have re-iterated both of them in our own ways.

1.3. We note that you have stated that you will provide repository information for your data at acceptance.

Our statements regarding data availability are “Yes - all data are fully available without restriction” and “All relevant data are within the manuscript and its Supporting Information files.” We didn’t say “ I will provide repository information at acceptance”. Hence, with due respect, I ask you to see and amend it ASAP.

1.4. We are unable to open your Supporting Information file [Echo Submitted to PLOS ONE.rar].

I will Re-upload the supporting Information file. It contains all the back ground data.

Remark: Diagnosis and classification, and operational definition part are revised. They were in table format and changed to plain write up.

2. Response to reviewers comments:

Dear reviewers, 

Thanks for the bold comments, complements and feedbacks you provided us. Here below are the responses for each points raised.

2.1. Review Comments to the Author and the responses

2.1.1. Reviewer 1:

2.1.1.1. Aim of the study: comments well accepted and edited. “Our aim is to describe the trans-thoracic echocardiography pattern of paediatric heart diseases and reasons of referral for echocardiography in children referred to Bahir Dar University Tibebe – Ghion Hospital and Adinas General Hospital.”

2.1.1.2. Title length: comment well accepted. The aim is well explained. “pattern”. The data were collected from two different hospitals with different levels and merging them might assume the two hospitals with same status. 

2.1.1.3. Punctuation and grammar errors: Well accepted and edited ASAP. 

2.1.1.4. Abstract section: comments well accepted and edited. Both the result and conclusion parts are re-iterated. In addition, key words are included. (see track changes and final manuscript version)

2.1.1.5. Introduction section: I agree with removal of paragraph – 4 on page three. It has nothing to do here at this part. ( references will nourish the methods part). Regarding paragraph – 2 on page – 4, in addition to describing the patterns, we aimed to describe the reasons that led children visit echocardiography evaluation. Hence, we would be very happy if it stays there as it is with minor rearrangement and modification. (see the revised manuscript)

2.1.1.6. End of introduction: we revised the objective statement and is now perfectly matching with your recommendation.

2.1.1.7. Methods: data collection methods and sources are included in the manuscript. In addition, who collected data and the decision on doubtful findings is also elaborated ASAP. Echo was done by pediatric cardiologist and in doubtful conditions, the adult cardiologists and radiologists will have a look at and suggest their recommendation. Any archived data with incomplete documentation was not included in the analysis. (see figure 1) Post intervention clients are included in the study as we are evaluating the echocardiography pattern of children with suspected heart disease. It reflects the magnitude of the intervention gap. 

2.1.1.8. Results: comment accepted. 

• For clarity, 3672 are all clients who underwent echocardiography evaluation at the two hospitals echocardiography laboratory over the four years period. The hospitals are located in an area where only 2 pediatric cardiologists are providing service. One in Gondar, North west Ethiopia and the other in Bahir Dar, where most of the Pediatric clients with suspected heart diseases are referred to for echocardiography evaluation. 

• Regarding the number of tables: Yes I agree with the reduction of number of tables. Table 2 and 3 are removed. It is revised in both the Manuscript and revised manuscript with track change.

• Table 1 Heading is revised in a way it was intended to …….. and converted to fig as it is larger in size to accommodate and insert it as table. The Table is intended to show the proportion of reasons of referral for echocardiography evaluation that end up with cardiac disorders/normal findings. 

2.1.1.9. Discussion: comments accepted and revised considering the inputs, comments and complements of the other reviewer too.

2.1.1.10. Thanks for all the valuable comments and complements.

2.1.2. Reviewer 2: 

2.1.2.1. Abstract:

2.1.2.1.1. The “indications” is replaced by “ reasons of referral for echocardiography”. The numbers might add up to more than the total number of clients. This is because, children might be referred for echocardiography with more than one reason. We put the clarification at the bottom of the table as remark.

2.1.2.1.2. Regarding “detection rate”, comments are well accepted. It was intended to show the proportion of reasons of referrals with positive cardiac findings. It is amended and revised that way.

2.1.2.1.3. Regarding “Univariate analysis”, once again comments are well taken. It is a descriptive analysis and amended ASAP.

2.1.2.2. Introduction: 

2.1.2.2.1. Too long introduction: comment is well accepted and shortened as recommended.

2.1.2.3. Methods:

2.1.2.3.1. Settings: comment accepted. The echocardiography models used for diagnostic assessment are described and mentioned. The capacity of the hospitals is elaborated with the extent of service and where the clients are getting services.

2.1.2.3.2. Analysis: comment accepted and it is descriptive analysis. Univariate analysis is used synonymously with descriptive analysis in some references and we found it that it is confusing. For clarity purpose and comforting the readers, we preferred to use the phrase “ descriptive analysis”.

2.1.2.3.3. “Ethical approval”: it is mentioned in the study design sub section. and the minute is already uploaded as a separate file during submission. In addition, as per your recommendation, I have added “Institutional Ethics Committee issued approval protocol number: 795/2023. This hospital-based retrospective observational study was approved by the Institutional Review Board of Bahir Dar University College of Medicine and Health sciences and was approved without comments.” In the disclosure part.

2.1.2.3.4. Regarding “ WHF 2012 Borderline RHD Criteria Vs. Stage A RHD”: thanks for the update. I included it in the operational definition and discussed it in the discussion part. Tried to show the difference and similarity.

2.1.2.4. Results: 

2.1.2.4.1. Figure 1: comment accepted and edited. it is edited and brought with a better one. In some of our tables and figures, the percent and numbers add up more than hundred percent and the total number of cases because a single client might have more than one cardiac lesion. For example Congenital Heart disease and acquired heart disease. So the readers are not expected to add up the numbers.

2.1.2.4.2. “Indications” is revised as “reasons of referral for echocardiography”. They can be single or multiple for a single client which in turn will increase both the precent and number more than 100% and total number of cases. Hence, we recommend the readers not to add up them rather to see the proportion of reasons for referral as it appears. Remarks are there at the bottom of the tables and figures.

2.1.2.4.3. Regarding “positive predictive value” : Our intention is not to show the “positive predictive value” rather the proportion of reasons of referral for echocardiography and the cardiac findings and to show which reasons are more common and linked with more cardiac abnormalities. Re-iterated.

2.1.2.4.4. Table 3 – 1 and Table 3 – 2 are revised and you are right they were Tables 4 – 1 and 4 – 2. 

2.1.2.4.5. Regarding the recommendations for use of univariate analysis and reporting associations: it is not our intention and descriptive analysis will not show association rather it is just a description.

2.1.2.4.6. Discussion, conclusion and recommendations: they all are re-visited and revised as per all the comments and complements given.

2.1.2.4.7. once again, thanks for the comments and complements.

---

## [Decision Letter · Decision Letter 1]

29 Jan 2024

PONE-D-23-28984R1Two – Dimensional and Doppler Trans-Thoracic Echocardiographic Patterns of suspected Pediatric Heart Diseases at Tibebe – Ghion Specialized Teaching Hospital and Adinas General Hospital, Bahir Dar, North West Ethiopia: – an experience from a LMIC.

PLOS ONE

Dear Dr. Gelaw,

Thank you for submitting your revised manuscript to PLOS ONE. After careful consideration, we feel that it has merit but does not fully meet PLOS ONE’s publication criteria as it currently stands. Therefore, we invite you to submit a revised version of the manuscript that addresses the points raised during the review process. In addition to comments from the reviewers, I have also added the below comments for major revision to your manuscript: 

Additional comments:

**Please add “cross-sectional study” in your study design. The terms mentioned represent the type of study (descriptive) and timeline of data collection (retrospective), not necessarily the study design.**

**The data was collected retrospectively, understandably, from the medical records. Why was verbal consent taken from the patients and how did the authors manage this? I would have expected them to mention they got waiver of consent from IRB.**

**The operational definition section, I know this was motivated by the reviewers’ comments on diagnosis of myocarditis. This diagnosis can be defined in the introduction or methods or discussion sections. Otherwise, it’s needless to give operational definition for all congenital or acquired diseases. It’s sufficient if they are contextually discussed where needed.**

**The statistical section is vague. It needs clarification on what analysis was used for what (e.g., numerical variables were presented as mean±SD or median (interquartile range) based on …, categorical variables were presented …, etc.)**

**Page 7, last paragraph, “Out of the 3, 647 pediatric clients with complete documentation and first-time visits; 2,273 children had one or more cardiac diseases on echocardiography”. Is it to mean one or more cardiac lesions? Cardiac disease is not the same as cardiac lesion.**

**Traditionally, table 1 is meant to describe baseline characteristics of study participants, not results. In addition, the classification as neonate, infant, etc. is not appropriate as the newborn is also included in the infant category. For clarity, it’s better to mention the age classification that the authors used (e.g., 0-28 days, 1-12 months, etc). in addition, this table contains a lot of cells with zero values which made the table so redundant. Some small numbers can be lumped together as “others”. This table needs to be contracted to no more than half a page; otherwise, it will be difficult for the readership to capture the message or even to read.**

**Figure 4 is redundant. The information has already been included in table 1.**

**Figure 5. The figure legends are given twice, once on the left side of the figure then at the bottom. In my opinion, this figure also doesn’t add much to the message. This figure also indicates that 15 neonates (0.004*3647) were diagnosed for acquired heart diseases. My question would be what were these acquired lesions in the newborns? What was the degree of certainty?**

**Discussion: I disagree with the last statement of the first paragraph. The authors need to emphasize their results in the context of the objective of the study. Judging the ability and skill of the referring physician was not the objective of the study. There are multitudes of reasons that physicians may refer patients, especially in setting where there are no guidelines. As a matter of fact, the authors mention that the most common reason for referral was detection of murmur, but we don’t in how many of these patients the murmur was substantiated or refuted by the cardiologists.**

** Page 12, last paragraph, statement about a US study doesn’t fit the context of your study and I recommend that you remove the statement and the reference.**

**Page 13, 1^st^ paragraph, mortality should be included as one of the reasons. Newborns with critical CHD may be misdiagnosed or may not even make it to the hospital which complicates your comparison. However, there is also an issue comparing it with a Cameroonian study and cite these reasons as explanations because the facility and infrastructure in Cameroon is not expected to be much different than that of yours. I recommend that you cite literature in a context that supports your arguments.**

**Page 13, last but not one paragraph, again the cited literatures are from low-income similar settings but when giving explanations, the authors say the difference is because of the setting, facilities, or maternal infections. This doesn’t make sense as compared settings are not that different.**

**The discussion section needs overhaul. My recommendation and personal preference:**

**Highlight main study findings in the 1^st^ paragraph,**

**Give brief contextual possible explanations for the findings,**

**Similarities and differences of methodology and findings with other similar studies, rather than citing publications randomly in the 3^rd^ paragraph,**

**Study limitations and constraints in the 4^th^ paragraph**

**The way forward and recommendation for future work in the last paragraph**

**Only relevant literature needs to be cited. These need to be selected based on critical appraisal of their methods and findings (that way the authors can limit the list of references).**

**Journals need to be written using their standard abbreviations and this needs to be applied throughout the reference list.**

**PLOS ONE does not copy edit manuscripts. Therefore, the authors need to attend to grammatical, syntax and punctuation errors: preferably seek help from native English speakers if possible.**

We look forward to receiving your revised manuscript.

Kind regards,

Endale Tefera

Academic Editor

PLOS ONE

Reviewers' comments:

Reviewer's Responses to Questions

**Comments to the Author**

1. If the authors have adequately addressed your comments raised in a previous round of review and you feel that this manuscript is now acceptable for publication, you may indicate that here to bypass the “Comments to the Author” section, enter your conflict of interest statement in the “Confidential to Editor” section, and submit your "Accept" recommendation.

Reviewer #1: All comments have been addressed

Reviewer #2: (No Response)

2. Is the manuscript technically sound, and do the data support the conclusions?

Reviewer #1: Yes

Reviewer #2: Partly

3. Has the statistical analysis been performed appropriately and rigorously? 

Reviewer #1: Yes

Reviewer #2: No

4. Have the authors made all data underlying the findings in their manuscript fully available?

Reviewer #1: Yes

Reviewer #2: Yes

5. Is the manuscript presented in an intelligible fashion and written in standard English?

Reviewer #1: Yes

Reviewer #2: Yes

6. Review Comments to the Author

Reviewer #1: The authors have addressed my comments very well. The language edits, the change in the format of the table and the complete revison that they have done in the discussion section have improved the manuscript significantly.

Reviewer #2: Dear Authors,

Thanks for revising your manuscript and addressing most of the comments.

Here below you find additional comments that were partly addressed or not addressed at all to make your article more readable.

- Please stress that your paper is only descriptive and mention that no further analytic analysis was made. This will make it clear for the end users as to the questions they might have while reading your paper. Please include this as the limitation of your study and if possible quote also the reasons for that like data availability issue etc.

- Please include the interquartile range for the median age of study participants on page 7.

- Please discuss the issue of making a diagnosis of myocarditis or cardiomyopathy in the face of an underlying heart disease or mention it as a limitation as you have a group patients who ended up with it.

- You still have many figures, can it be reduced? Like what is the main difference between Stable 1 and Figure 5. They have related information. Please drop one of them

- Figure 3 is challenging for end users i.e. proportions vs clinical symptoms. However, as the authors opted out for only descriptive report, then it can be kept as it is. Please move it to supplemental section.

- Please make the recommendation short and precise and based on your study findings. It is long as it is now.

- Please address the missing capitalization and syntax issues in the paper one more time.

7. PLOS authors have the option to publish the peer review history of their article (what does this mean?). If published, this will include your full peer review and any attached files.

Reviewer #1: **Yes: **Atnafu Mekonnen Tekleab

Reviewer #2: No

---

## [Author Response · Author response to Decision Letter 1]

9 Feb 2024

1. Response to Academic Editor: Thank you for the critical and valuable feedbacks and comments that nourish our work significantly.

1.1. Please add “cross-sectional study” in your study design. The terms mentioned represent the type of study (descriptive) and timeline of data collection (retrospective), not necessarily the study design.

1.1.1. Response: comment well accepted and added to the methods part.

1.2. The data was collected retrospectively, understandably, from the medical records. Why was verbal consent taken from the patients and how did the authors manage this? I would have expected them to mention they got waiver of consent from IRB.

1.2.1. Response: comment well accepted and revised (see previous review comment)

1.3. The operational definition section, I know this was motivated by the reviewers’ comments on diagnosis of myocarditis. This diagnosis can be defined in the introduction or methods or discussion sections. Otherwise, it’s needless to give operational definition for all congenital or acquired diseases. It’s sufficient if they are contextually discussed where needed.

1.3.1. Response: Comment accepted and removed and inserted ASAP.

1.4. The statistical section is vague. It needs clarification on what analysis was used for what (e.g., numerical variables were presented as mean±SD or median (interquartile range) based on …, categorical variables were presented …, etc.)

1.4.1. Response: comment well accepted and re-visited. 

1.5. Page 7, last paragraph, “Out of the 3, 647 pediatric clients with complete documentation and first-time visits; 2,273 children had one or more cardiac diseases on echocardiography”. Is it to mean one or more cardiac lesions? Cardiac disease is not the same as cardiac lesion.

1.5.1. Response: It is to mean cardiac lesion. Revised as per the recommendation.

1.6. Traditionally, table 1 is meant to describe baseline characteristics of study participants, not results. In addition, the classification as neonate, infant, etc. is not appropriate as the newborn is also included in the infant category. For clarity, it’s better to mention the age classification that the authors used (e.g., 0-28 days, 1-12 months, etc). in addition, this table contains a lot of cells with zero values which made the table so redundant. Some small numbers can be lumped together as “others”. This table needs to be contracted to no more than half a page; otherwise, it will be difficult for the readership to capture the message or even to read.

1.6.1. Response 1: the baseline characteristics of study participants in our study are age and gender of children assessed. The reason we omit this one is that they are reflected in the narration, table and figures parts in one or another way. 

1.6.2. Response 2: the comment on the classification of age of children assessed is well accepted and corrected as per the recommendation provided.

1.6.3. Response 3: comment regarding the size of the table is well accepted and contracted as much as possible.

1.7. Figure 4 is redundant. The information has already been included in table 1.

1.7.1. Response: comment accepted and the figure is removed.

1.8. Figure 5. The figure legends are given twice, once on the left side of the figure then at the bottom. In my opinion, this figure also doesn’t add much to the message. This figure also indicates that 15 neonates (0.004*3647) were diagnosed for acquired heart diseases. My question would be what were these acquired lesions in the newborns? What was the degree of certainty?

1.8.1. Response 1: The figure legends are duplicated. We removed the legend at the bottom. 

1.8.2. Response 2: The figure’s added value in nourishing our work: It’s added value is not significant as the content in it is already there in table 1. We agree to remove it from the manuscript.

1.8.3. Response 3: The number of neonates with acquired heart disease diagnosis: They are only 2 (0.004 *501 = 2). It is from total neonates screened not from total children screened. The acquired heart lesions were Cardiomyopathy associated with perinatal asphyxia and myocarditis. We used operational definition for diagnosis of Cardiomyopathy and myocarditis.

1.9. Discussion: I disagree with the last statement of the first paragraph. The authors need to emphasize their results in the context of the objective of the study. Judging the ability and skill of the referring physician was not the objective of the study. There are multitudes of reasons that physicians may refer patients, especially in setting where there are no guidelines. As a matter of fact, the authors mention that the most common reason for referral was detection of murmur, but we don’t in how many of these patients the murmur was substantiated or refuted by the cardiologists.

1.9.1. Response 1: comment well accepted and we believe that it has extended out of the objective and it cannot provide a clue for such a judgment to be concluded. It is revised.

1.9.2. Response 2: the most common reason of referral: it only provides the documented reason of referral for echocardiography given by any health care provider. As it is a retrospective analysis of documentation at echo lab., it was not possible to get a documentation that substantiated or refuted the finding based on further clinical evaluation by cardiologists and our objective was not to substantiate or refute it. We revised this part of the discussion in such a way to be clear for readers.

1.10. Page 12, last paragraph, statement about a US study doesn’t fit the context of your study and I recommend that you remove the statement and the reference.

1.10.1. Response: comment well accepted and it is removed.

1.11. Page 13, 1st paragraph, mortality should be included as one of the reasons. Newborns with critical CHD may be misdiagnosed or may not even make it to the hospital which complicates your comparison. However, there is also an issue comparing it with a Cameroonian study and cite these reasons as explanations because the facility and infrastructure in Cameroon is not expected to be much different than that of yours. I recommend that you cite literature in a context that supports your arguments.

1.11.1. Response: comment well accepted. Reasons for higher proportion of children in the age group from one month to twelve months are updated as recommended. Evidences that doesn’t support our argument are removed. 

1.12. Page 13, last but not one paragraph, again the cited literatures are from low-income similar settings but when giving explanations, the authors say the difference is because of the setting, facilities, or maternal infections. This doesn’t make sense as compared settings are not that different.

1.12.1. Response: evidences that are not substantiating our discussion are removed.

1.13. The discussion section needs overhaul. My recommendation and personal preference:

1.13.1. Response: comment well accepted and revised as much as possible to suit the recommendations provided.

1.14. Journals need to be written using their standard abbreviations and this needs to be applied throughout the reference list.

1.14.1. Response: comment accepted and done ASAP.

1.15. PLOS ONE does not copy edit manuscripts. Therefore, the authors need to attend to grammatical, syntax and punctuation errors: preferably seek help from native English speakers if possible.

1.15.1. Response : comment accepted.

2. Response to reviewer 2: thank you very much for tirelessly evaluating our work tirelessly and commenting constructively.

2.1. Please stress that your paper is only descriptive and mention that no further analytic analysis was made. This will make it clear for the end users as to the questions they might have while reading your paper. Please include this as the limitation of your study and if possible quote also the reasons for that like data availability issue etc.

2.1.1. Response: comment accepted well and revised as per the recommendation.

2.2. Please include the interquartile range for the median age of study participants on page 7.

2.2.1. Response: comment accepted and IQR is included.

2.3. Please discuss the issue of making a diagnosis of myocarditis or cardiomyopathy in the face of an underlying heart disease or mention it as a limitation as you have a group patients who ended up with it.

2.3.1. Response: comment accepted and revised. 

2.4. You still have many figures, can it be reduced? Like what is the main difference between Stable 1 and Figure 5. They have related information. Please drop one of them.

2.4.1. Response: comment accepted. Stable 1, figure 4 and figure 5 are removed as they can be reflected in the other tables and figures.

2.5. Figure 3 is challenging for end users i.e. proportions vs clinical symptoms. However, as the authors opted out for only descriptive report, then it can be kept as it is. Please move it to supplemental section.

2.5.1. Response: the intention of presenting figure three here is to show the proportion of children identified with heart disease against the presenting complaints/reasons of referral for echocardiography. It is only for descriptive report. We pushed it to supplemental section.

2.6. Please make the recommendation short and precise and based on your study findings. It is long as it is now.

2.6.1. Response: comment well accepted and revised.

2.7. Please address the missing capitalization and syntax issues in the paper one more time.

2.7.1. Response: comment accepted and revised.

Reviewer 1 had no comments or feedbacks in this session.

---

## [Decision Letter · Decision Letter 2]

23 Feb 2024

Two – Dimensional and Doppler Trans-Thoracic Echocardiographic Patterns of suspected Pediatric Heart Diseases at Tibebe – Ghion Specialized Teaching Hospital and Adinas General Hospital, Bahir Dar, North West Ethiopia: – an experience from a LMIC.

PONE-D-23-28984R2

Dear Dr. Gelaw

We’re pleased to inform you that your manuscript has been judged scientifically suitable for publication and will be formally accepted for publication once it meets all outstanding technical requirements.

Kind regards,

Endale Tefera

Academic Editor

PLOS ONE

Additional Editor Comments (optional):

Reviewers' comments:

Reviewer's Responses to Questions

**Comments to the Author**

1. If the authors have adequately addressed your comments raised in a previous round of review and you feel that this manuscript is now acceptable for publication, you may indicate that here to bypass the “Comments to the Author” section, enter your conflict of interest statement in the “Confidential to Editor” section, and submit your "Accept" recommendation.

Reviewer #2: All comments have been addressed

2. Is the manuscript technically sound, and do the data support the conclusions?

Reviewer #2: Yes

3. Has the statistical analysis been performed appropriately and rigorously? 

Reviewer #2: Yes

4. Have the authors made all data underlying the findings in their manuscript fully available?

Reviewer #2: Yes

5. Is the manuscript presented in an intelligible fashion and written in standard English?

Reviewer #2: Yes

6. Review Comments to the Author

Reviewer #2: Dear Authors

Thanks for addressing the concerns. Please do one more time language edits. Otherwise comments are well addressed.

7. PLOS authors have the option to publish the peer review history of their article (what does this mean?). If published, this will include your full peer review and any attached files.

Reviewer #2: **Yes: **Henok Tadele

---

## [Editor Report · Acceptance letter]

29 Feb 2024

PONE-D-23-28984R2 

PLOS ONE

Dear Dr. Gelaw, 

I'm pleased to inform you that your manuscript has been deemed suitable for publication in PLOS ONE. Congratulations! Your manuscript is now being handed over to our production team.

Kind regards, 

on behalf of

Dr. Endale Tefera 

Academic Editor

PLOS ONE